# Accreditation and Certification: Do They Improve Hospital Financial and Quality Performance?

**DOI:** 10.3390/healthcare9070887

**Published:** 2021-07-14

**Authors:** Matthew Brooks, Brad M. Beauvais, Clemens Scott Kruse, Lawrence Fulton, Michael Mileski, Zo Ramamonjiarivelo, Ramalingam Shanmugam, Cristian Lieneck

**Affiliations:** School of Health Administration, Texas State University, Encino Hall, Room 250A, 601 University Drive, San Marcos, TX 78666, USA; mbrooks@txstate.edu (M.B.); Scottkruse@txstate.edu (C.S.K.); Lf25@txstate.edu (L.F.); mileski@txstate.edu (M.M.); zhr3@txstate.edu (Z.R.); Shanmugam@txstate.edu (R.S.); clieneck@txstate.edu (C.L.)

**Keywords:** CAHME, ACHE, program accreditation, professional affiliation, healthcare outcomes, financial performance

## Abstract

The relationship between healthcare organizational accreditation and their leaders’ professional certification in healthcare management is of specific interest to institutions of higher education and individuals in the healthcare management field. Since academic program accreditation is one piece of evidence of high-quality education, and since professional certification is an attestation to the knowledge, skills, and abilities of those who are certified, we expect alumni who graduated from accredited programs and obtained professional certification to have a positive impact on the organizations that they lead, compared with alumni who did not graduate from accredited programs and who did not obtain professional certification. The authors’ analysis examined the impact of hiring graduates from higher education programs that held external accreditation from the Commission on Accreditation of Healthcare Management Education (CAHME). Graduates’ affiliation with the American College of Healthcare Executives (ACHE) professional healthcare leadership organization was also assessed as an independent variable. Study outcomes focused on these graduates’ respective healthcare organization’s performance measures (cost, quality, and access) to assess the researchers’ inquiry into the perceived value of a CAHME-accredited graduate degree in healthcare administration and a professional ACHE affiliation. The results from this study found no effect of CAHME accreditation or ACHE affiliation on healthcare organization performance outcomes. The study findings support the need for future research surrounding healthcare administration professional graduate degree program characteristics and leader development affiliations, as perceived by various industry stakeholders.

## 1. Introduction

Nearly two decades ago, the Institute of Medicine (IOM) published *Crossing the Quality Chasm* in response to the organization’s prior groundbreaking publication demonstrating the effects of human error and related issues in the U.S. healthcare industry. Many reports of medical errors and related industry waste and abuse claimed similar systematic issues, but it was the IOM’s efforts of the U.S. healthcare system reporting on its own industry shortfalls that was most impactful. Specifically, it characterized the provision of care in the United States by claiming:

The American health care delivery system is in need of fundamental change. Many patients, doctors, nurses, and health care leaders are concerned that the care delivered is not, essentially, the care we should receive. The frustration levels of both patients and clinicians have probably never been higher. Yet the problems remain. Health care today harms too frequently and routinely fails to deliver its potential benefits [1,2,3].

To cross the identified chasm, the IOM provides valuable recommendations to right the ship for the U.S. healthcare industry and its multiple stakeholders. However, the report also claims that what is most concerning is “[an] absence of real progress toward restructuring health care systems to address both quality and cost concerns…” [3]. Unfortunately, and despite the best efforts of policy makers and the passage of the Affordable Care Act (ACA) in 2010, progress towards improved care outcomes and cost reduction has been frustratingly slow. As of 2019, the United States spends more on healthcare than any other developed nation, yet has the lowest life expectancy, the highest chronic disease burden, the highest number of hospitalizations from preventable causes, and the highest rate of avoidable deaths [4]. Therefore, it is time to look beyond the clinical and administrative functions that have been a primary focus of recent reform efforts and focus on the leaders who guide these healthcare organizations—specifically, how these leaders are prepared academically and if they are affiliated with follow-on professional development to meet contemporary industry challenges.

The value of external accreditation for graduate degree programs and certifications offered by various professional stakeholder organizations continues to be an area of specific interest to academic program administrators and especially to current healthcare administrators leading these organizations toward improved, measurable outcomes [5]. Beyond leadership preparations, hospital accreditation itself may not demonstrate significantly lower mortality rates or patient experience scores [6,7]. In fact, hospital accreditation has mixed study results for financial, quality, and program assessment measures [8]. Additionally, the entire process of ranking and evaluating accredited graduate health administration programs is quite subjective, although more empirical methods have been suggested [9]. This is the first study to assess the linkage between graduate healthcare management program accreditation status, graduate professional association development engagement, and healthcare facility outcomes.

While some types of professional certifications have improved program graduate marketability and the hiring of graduating healthcare management students [10], the research team found little evidence to support the notion that professional certifications have led to better healthcare outcomes for industry stakeholders. Notably, from the field of nursing, a systematic review of studies found inconsistent and contradictory evidence regarding the relationships between outcome and certifications [11].

### 1.1. Accreditation and Professional Certification Costs

One driving reason that this study was undertaken was due to the cost incurred to maintain accreditation and personal certification. Accreditation cost studies are difficult to find in the literature; however, a dated study has shown that at least one accrediting body is cognizant of the price structure and accordingly extended accreditation lengths, reduced the number of visits, and leveraged teleconferencing [12]. The cost for maintaining CAHME accreditation for an average academic program is USD 4900 per year [13]. Further, CAHME accreditation requires membership in the Association of University Programs in Healthcare Administration (AUPHA) at an additional annual membership cost based on the type of program and number of students. A small undergraduate program with under 500 students could pay as little as USD 2655 while a larger graduate program with over 500 students will pay USD 11,457 [14].

The annual cost for a Fellow in ACHE (FACHE) to maintain their affiliation is USD 350, with an additional USD 200 for mandatory recertification every three years. Further, to be recertified, a FACHE must achieve 36 continuing education hours, of which 12 must be face-to-face education, within a three-year timeframe. For many, this entails attending the Annual Congress in Chicago, IL in March at a registration cost of USD 1449 for ACHE members and USD1749 for nonmembers, plus additional travel-related expenses [15]. The typical net cost for three years will then be around USD 2700 or around USD 900 per annum not including travel.

### 1.2. Accreditation and Professional Certification Marketing

The CAHME, website states: “Students from CAHME accredited programs can access more job opportunities. Many employers will interview only those students who graduate from a CAHME-accredited program” [13]. The ACHE webpage states: “When our members excel, healthcare as a whole excels too. When they advance, patients and communities benefit alike” [15]. These two organizations imply that senior leaders in healthcare should be educated by CAHME-accredited programs and be affiliated with ACHE. This would also further imply that organizations who hired leaders with these qualifications should have better patient and financial outcomes compared with leaders without these qualifications.

Given these statements and the relative dearth of extant literature on the topic, we investigate the effects of accreditation and certification on hospital financial and Hospital Value-Based Purchasing (HVBP) Total Performance Scores (TPS) for facilities in Texas with the assumption that these metrics should be higher when more of the top management team were educated in CAHME-certified programs and/or have FACHE credentials. This study focuses on Master’s in Healthcare Management degrees that are accredited by CAHME and the professional affiliations of faculty, students, and alumni in ACHE.

### 1.3. Research Question

While costs for accreditation and professional certification are non-trivial, the institutional forces pushing it are compelling. This leads us to ask: does an accredited Master of Healthcare Administration (MHA) degree and follow-on professional development improve organizational cost, quality, and access outcomes at the hospital level? Scoping this question, the research team investigated the association between hospital net operating profit margin with hospital characteristics (controls), population characteristics (controls), and the senior administrative leadership team credentials for hospitals in Texas. Secondarily, the Hospital Value-Based Purchasing (HVBP) Total Performance Score (TPS) was also investigated as a function of these same explanatory variable groupings. 

### 1.4. Significance

To the research team’s knowledge, this is the first study of its type to evaluate the contribution of professional development leadership credentials, specifically the Fellow in the American College of Healthcare Executives (FACHE), as well as the graduate program accreditation status with CAHME as predictors of hospital quality and financial performance. Similar to the critical analysis of the clinical pathways of care delivery that have occurred over the past several decades, an assessment of hospital leader development should be measurable as well as evidence-based, and should extend to the education, accreditation, and professional certification programs that are prevalent in the industry today. Most importantly, outcomes of accreditation and professional certification should be measured and used to guide continued program development for higher education and individual professional development alike.

The study proceeds as follows. First, we examine the relevant literature and then consider a theoretical model. Next, the data, variables, methods, models, and results are evaluated, with all analyses and additional investigations freely available at: https://rpubs.com/R-Minator/AC. Finally, the results along with the associated discussion and conclusions follow.

## 2. Literature

There have been many studies to examine the effectiveness of both academic program accreditation and professional organization affiliations. In most cases, these have focused on specific clinical medical education and medical licensure requirements at the State levels. A study by Davis and Ringsted in 2006 examined how accreditation standards in medical education contribute to quality outcomes [16]. The authors concluded that the process is focused on the specific aspects of the delivery of the education to meet an established set of standards, but there was no evidence that meeting these standards in fact had a measurable impact on the quality of care delivered by graduates. In fact, the key take-away was that accreditation standards have moved in a competency-based evaluation direction; therefore, the evidence of the benefit of the accreditation needs to be moved to competency-based outcomes as well [15]. This is the direction that CAHME took when adopting its competency model standard.

In an article by Barzansky et al. [17], the authors again examined accreditation in medical education, and the benefits that accreditation had for quality improvement. This study focused on the process from a continuous quality improvement (CQI) perspective. The authors concluded that periodic accreditation visits, often using lag data, were not effective in demonstrating a CQI outcome for the academic programs. In many cases, an accreditation visit occurs between every 6 and 10 years. This results in the evaluation of the program using data which are 1–2 years old and is not able to make any real-time impact on CQI outcome. In the case of CAHME, site visits occur on average every 7 years for reaccreditation [17]. Information used at this site visit also ranges in the one to two-year-old range.

A study by Anderson and Garman [18] examined the impact that CAHME accreditation had on three specific outcomes: applicant quality, program selectivity, and mean starting salaries. This analysis aligns with the specifics of healthcare management education, yet falls short of what the true outcomes of the accreditation are from a delivery of healthcare concept. As with previous studies in the area, the focus continues to be associating a positive outcome from accreditation to be based solely on the academic experience. The authors in this study found that the longer the tenure of CAHME accreditation, the higher the quality of the applicant pool and the more selective the program could be in admissions. This is a competitive environment issue, and a basic supply and demand outcome. The question remains, and what has not been identified in the literature is the link that programmatic accreditation must improve healthcare delivery outcomes [18].

The literature review for this analysis shifts now away from academic program accreditation to specific professional association membership in healthcare management, especially in ACHE and achieving the FACHE credentials evidence to improve healthcare outcomes. A paper published by Bowen and Hahn [19] outlines the specifics of the FACHE process. The basic requirements are a CAHME-accredited Master’s degree, at least 5 years of healthcare management experience, and successful completion of the Board of Governor’s Exam (BOG). The authors conclude that achieving this credential demonstrates a commitment to the field of healthcare and establishes a measure of leader competency [19]. In a study by Kaliq and Waltson [20], the authors examined a cross-sectional data set of 582 Chief Executive Officers (CEOs) in U.S. healthcare facilities that were either members or Fellows in ACHE. Their analysis showed that 162 were ACHE members, and 272 (47%) were current fellows. Again, this study examined the likelihood that being a CEO in healthcare was associated with ACHE affiliation but did not demonstrate that this is associated with improvement in the quality of healthcare delivered [20].

The relevant literature includes additional articles that focus on the benefits of professional organization affiliation in general aspects, and always from a specific organization. In a 2010 article by Mata, Latham, and Ransomed [21], the authors examine professional organization membership and its corresponding attendance at national meetings for students and new professionals. This study concludes that the benefit to this younger population is categorized by opportunities for career development, skill-building, and networking. An article by Escoffery, Kenzig, and Hyden [22] reaches the same conclusion regarding the benefit of professional affiliations in healthcare. They examine multiple professional organizations, including ACHE, and conclude that the primary outcome is networking. A consistent theme throughout the literature demonstrates that the benefit of affiliation is networking. This would be the outcome of all professional organizations, but the link between establishing a professional network and corresponding improvement in healthcare outcomes remains unanalyzed.

One article by Gerard [23] takes a slightly different approach to analyzing professional affiliations. This author explores the dangers of healthcare management becoming too professionalized. He reiterates what has already been discussed, namely that one reason the field has moved in this direction is a response to the uncertainty in the field. The author concludes that the professionalization that comes from professional organization affiliations in healthcare management is potentially preventing a broader approach to the issues and prevents the field from advancing to a more patient experience approach to healthcare delivery [23].

## 3. Theoretical Model

Considered in the context of the literature, we considered that the most applicable theoretical model to this area of research is isomorphism based on institutional theory. Isomorphism refers to a process whereby organizations exposed to the same institutional environments tend to adopt similar characteristics and practice. The model of isomorphism is divided into competitive and institutional isomorphism [24]. There is a competitive aspect to accreditation and affiliation, one supported by a systematic review of the healthcare literature [25]. In many cases, academic program leadership seek accreditation to maintain a competitive position [26]. When addressing accreditation for business schools, Zhao and Ferran stated that “accreditation is no longer a luxury but a requirement for business schools.” [27].

The theoretical focus of this study, institutional isomorphism, includes three categories: coercive, mimetic, and normative [24]. Coercive focuses on the political influences and the need for legitimacy that are often associated with cultural expectations. In some cases, this is driven by mandates or regulations, but often it is the expectation that the organization needs to be associated with these organizations for legitimacy [24]. In the case of healthcare, many scholarships and fellowships require that the individual graduate from a CAHME-accredited program [26], and many healthcare industry job positions, require FACHE credentials.

In reviewing the mimetic aspect of the institutional isomorphism theoretical model, we see that organizations seek these affiliations to help offset uncertainty in the marketplace. This theory shows that organizations tend to model themselves after similar, or perceived successful, organizations to facilitate a sense of legitimacy [24]. This modeling demonstrates a mimetic process instead of being based on any specific and documented outcome or efficiency. Mimetic theory also shows the tendency towards homogeneity in organizational structure and processes, as we often see that new organizations are modeled after existing ones [28,29,30]. Both CAHME and ACHE communicate that the benefits or association are a specific set of consistent standards across their affiliated organizations or individuals. In many cases, these arguments are plausible and have value. On the other hand, we must consider that “groupthink” may affect the diversity of ideas and approaches to solutions. This leads us to question if mimetic theory implies that CAHME accreditation and ACHE affiliation lead to organizational homogeneity [24].

The normative aspect of institutional isomorphism focuses on the need for professionalization and the desire to establish cognitive occupational standards. This certainly applies in healthcare, where credentialing and a specific set of occupational standards are required to be hired and perform in the industry [25]. One aspect of the normative isomorphism is the belief that formal education delivered by faculty with specific qualifications is the only successful way to achieve educational goals and that competency models are the single best way to ensure that students acquire the knowledge necessary to succeed. Therefore, we would expect healthcare executives who graduated from CAHME-accredited programs and those who acquired ACHE Fellowship to have the same knowledge base, managerial skills, and behaviors to successfully lead healthcare organizations in terms of quality care and financial performance. A second aspect of professionalization is of building a network of peers and institutions, which often leads to a way to efficiently and effectively diffuse information. There are benefits to building a network, and having professional relationships across every industry, and this is especially true for healthcare [24]. These associations have additional benefits in recruiting and hiring staff, as well as in the marketing and recognition of both individuals and organizations. While these theoretical effects of institutional isomorphism will be realized in many industries, the absence of evidence-based improvements in organizational efficiency or outcomes requires work by those involved in education. Institutional theory highlights why an academic organization seeks accreditation, and why individuals seek and obtain professional organization affiliation. What it is does not demonstrate is whether accreditation and professional certification have a positive relationship with organizational outcomes that can be measured in terms of the Iron Triangle [2].

Therefore, based on our study and the application of Institutional Theory, we hypothesize that:

**Hypothesis** **1.**
*Hospitals who employ healthcare executives who graduated from CAHME-accredited programs and are ACHE-certified demonstrate higher financial performance compared with hospitals who employ healthcare executives who did not graduate from CAHME-accredited programs and are not ACHE-certified.*


**Hypothesis** **2.**
*Hospitals who employ health are executives from CAHME-accredited programs and are ACHE-certified provide higher-quality care compared with hospitals who employ healthcare executives who did not graduate from CAHME-accredited programs and are not ACHE-certified.*


## 4. Materials and Methods

### 4.1. Data

Data from 2019 regarding healthcare organizational leaders’ MHA degree programs and ACHE members statuses were manually extracted from LinkedIn and individual hospital public-facing websites. When data were unavailable, “unknown” status was used. Facility data were obtained from Definitive Healthcare [31]. The Definitive Healthcare database compiles United States hospital data sources including the Medicare Cost Reports, commercial claims data, Medicare Standard Analytics Files, Centers for Medicare and Medicaid Services (CMS) Hospital Compare, among others [31]. The cost report contains provider information such as facility characteristics, utilization data, cost, and charges by cost center (in total and for Medicare), Medicare settlement data, and financial statement data.

#### Unit of Analysis, Location, and Variables

The unit of analysis for this study was the hospital. The location was Texas. Inclusion criteria were short-term acute-care hospitals participating in CMS Hospital Value-Based Purchasing (HVBP). Table 1 provides the variables and their associated definitions. All variables remained untransformed during the analysis for readability, and residual analysis supported this decision.

### 4.2. Dependent Variables

Two dependent variables were of interest: value-based purchasing TPS to measure healthcare quality and net operating profit margin to measure financial performance. Value-based purchasing TPS is a measure designed by the Centers for Medicare and Medicaid Services (CMS). This measure is a linear combination of efficiency/cost reduction, clinical care, patient experience, and safety metrics [32]. Net operating profit margin is defined as: (Net Patient Revenue—Total Operating Expenses)/Net Patient Revenue. It is an important measure of how well an organization enhances revenue and controls its costs in its core business operations, with positive values indicating earnings and negative values indicating losses [31]. 

### 4.3. Independent Variables 

Five groups of variables served as controls. These groups included demographics, economic variables, health status, case complexity, and leader certification. Demographic control variables, based on the hospitals’ counties, included population, population density, racial distribution, and the proportion of population over 65 years old. Economic variables from the Bureau of Labor Statistics included estimates for unemployment rate (2019 data available) and household income (2018 data). Health status controls included the proportion of the population (by county) with obesity, cancer, Chronic Obstructive Pulmonary Disorder (COPD), diabetes, and heart failure. One variable, the average Case Mix Index (CMI) in the county, served as information about the hospitals’ case complexity. To account for competition, acute beds in the county was considered as a possible variable, but it was highly correlated with the 2020 population estimate for the county (*r* = 0.99) and thus omitted due to collinearity and the associated estimation issues (e.g., unreliable parameter estimates, inflated standard errors, etc.). Leadership ACHE certifications and leader graduate program CAHME accreditation was the final group. The FACHE status for the Chief Executive Officer (CEO), Chief Operating Officer (COO), and Chief Financial Officer (CFO) were tracked as trichotomous variables: No, Yes, Unknown. Further, the leaders’ graduate program CAHME status was also tracked in the same fashion. Feature engineering from these variables produced three dichotomous variables: any of the three leaders with known FACHE status, any of the three leader graduate programs with known CAHME accreditation, any leader FACHE or leader program CAHME certification. The number of leaders with known certifications was also tallied for FACHE, CAHME, and both FACHE and CAHME. 

### 4.4. Models 

The models of interest for both research questions follow Equation (1). In Equation (1), each dependent variable (*y*) is evaluated as a function (*f*) of demographic variables (*d*), economic variables (*e*), health status variables (*h*), competition/complexity variables (*c*), leadership credentials/university credentials (*x*), and error (*e*).
*y = f (d, e, h, c, x, e)*(1)

To prevent overfitting, lasso, ridge, and elastic net regression analyses were built using leave-one-out cross validation (LOOCV). This technique fits *n* (in this case 199) models, leaving one observation out each time for forecasting. Then, the forecasting metrics (i.e., root mean squared error, coefficient of determination) are compiled on the set of forecasts. Lasso and ridge provide L1 (absolute deviation) and L2 (squared deviation) penalty functions to regression models (respectively) in order to produce parsimonious models without overfitting. Elastic net combines both the L1 and L2 penalty functions via a weighting parameter. These techniques are essential in that they identify the variables in the model that are likely to be important. Asymptotically, the Akaike Information Criterion produces the same model as LOOCV. The technique producing the smallest predicted root mean squared error (RMSE) from parameter-tuning of the penalty functions was chosen to build the final ordinary least squares regression model that is unbiased. 

### 4.5. Software 

Non-online data were maintained in Microsoft Excel [33] and .csv files, while all analysis was performed in the R statistical software program [34]. The integrated development environment, R Studio, provided the platform for programming. Elastic net [35], lasso regression [36], and Tikhonov (ridge) regression [37] were performed in the glmnet package in R [38]. All analyses are freely available here: https://rpubs.com/R-Minator/AC.

## 5. Analyses and Results

### 5.1. Dependent Variables Descriptive Statistics

Table 2 provides the descriptive statistics for the quantitative and dichotomous variables. The “average” hospital had an operating profit margin of −0.09, a TPS score of 36.06, and served a county population of 1.2 million. The descriptive results for these variables indicate that the average hospital in Texas is not profitable and that quality assessment is low.

### 5.2. Independent Variable Descriptive Statistics

According to Table 2, the Native American population served by each hospital was relatively small (0.3%), while the Hispanic American and African American populations were significant (37.5% and 11.1% on average, respectively). The average population density was 22.3 K per square meter. The unemployment rate for each county was around 3.57%, while the average income was USD 59,678. Obesity, cancer, COPD, diabetes, and heart failure rates averaged 31.1, 7.6, 11.4, 29.5, and 15.9 per 100,000 population, respectively. The average CMI was 1.737. Around 76% of the hospital leaders were identified as having FACHE certifications or having come from a CAHME-accredited graduate program, with 50.3% of any administrative leadership team having at least one member as an FACHE and 49.2% coming from a CAHME-accredited program. 

The two dependent variables, operating profit margin and TPS, were uncorrelated (see Figure 1). TPS was approximately normally distributed with the single outlier removed (Shapiro–Wilk normality test W = 0.987, *p* = 0.075). The operating profit margin was not normally distributed, even with the removal of outliers and after optimal Box–Cox transformations and location adjustment.

Figure 2 shows the distribution of questions regarding FACHE and CAHME status of leaders (CEOs, CFOs, and COOs). For COOs, many did not provide their graduate institutions or FACHE status on the hospital websites or LinkedIn. Other missing data were retained, as the absence of responses reflected a phenomenon that might be interesting when predicting TPS and profit performance: marketing and transparency.

For CEOs, 33% were from CAHME-accredited universities, with 40% advertising the FACHE status. CFOs were unlikely to be from CAHME-accredited universities or be FACHE (7% for both). COOs were less likely to identify being either from CAHME-accredited universities (19%) or FACHEs (11%). Figure 3 depicts the CEO, CFO, and COO tenure. It was much more difficult to find the tenure status of COOs (55% unknown). The modal tenure for CEOs as well as CFOs was between 0 and 3 years (52% and 33%, respectively). For variable selection, all quantitative variables were scaled as part of pre-processing. In regularized regression (e.g., lasso), scale invariance does not exist. 

## 6. Models

### 6.1. Operating Profit Margin

For the first research question, the research team evaluated the operating profit margin as a function of the five variable groupings using three separate models and LOOCV to estimate a final model. The best model after tuning was the elastic net, which predicted only an *R*^2^ = 0.014 of the sum of squares variability and exhibited a large RMSE of 0.994. This model selected only CEO tenure between 7 and 9 years to be in the model (parameter estimate = −0.074). The lasso regression proffered an intercept only model (i.e., the null). Ridge regression suggested a negative adjusted R^2^ since it retains many more variables and shrinks coefficients towards zero. The largest single variable coefficient was small (−0.04) and associated with CFO tenure between 7 and 9 years. Running an analysis of variance for scaled profit margin as a function of CFO tenure alone resulted in a model that was statistically significant (F (_1,197_) = 4.796, *p* = 0.029) but with negligible effect size, η^2^ = 0.024. No variable grouping helped to predict operating profit margin in a reasonable fashion. Therefore, Hypothesis 1 was not supported.

### 6.2. Total Performance Score

The second research question involved modeling TPS as a function of the variable groupings. In the case of TPS, elastic net and lasso models produced nearly identical RMSE (0.943 versus 0.941, respectively). The elastic net model had the best adjusted R^2^, 0.098, with the lasso being similar (0.095). The adjusted R^2^ for the ridge regression was negative due to the penalty of including more variables. The recommended coefficients from the elastic net and lasso models were COO CAHME status “unknown”, CFO CAHME status “yes”, COO FACHE status “unknown”, CEO tenure (greater than 10 years), any certification of the leaders or their graduate programs (FACHE or CAHME), Native American status, African American/Black status, population density, and unemployment. These coefficients were then used in an initial OLS regression and statistically significant variables were retained for the final model. Only three variables that were statistically significant with evidence of predictive value remained in the final model: Native American population status, population density, and the presence or absence of any leader or leader program certification. The effect size was again nominal (*R*^2^ = 0.092). Table 3 is the coefficient table. A Shapiro–Wilk test failed to reject the null assumption of normality (W = 0.991, *p* = 0.253). These results indicate that Hypothesis 2 was not supported.

The results indicate that TPS declines as the proportion of the Native American population increases. Increases in population density are associated with better TPS scores. Finally, the presence of any individual or program certification has a negative effect on TPS scores. The effect sizes of the entire model and individual coefficients are small. In total, the results speak little to the effect of individual or program certifications on the TPS or profit margin.

## 7. Discussion

This analysis looked at the association between CAHME-accredited program preparation and ACHE Fellow professional affiliation of key healthcare leaders and the demonstrated outcomes of the facilities in which they work. Although both CAHME and ACHE actively market their importance and impact on the industry, little quantitative research into either entities’ direct influence on healthcare cost, quality, or access can be found in the literature.

This study showed no effect of accreditation and professional certification on hospital performance metrics. The lack of finding was unexpected, as we hypothesized that there would be visible value in one or both of these variables. While this is a preliminary study, the lack of findings should motivate accreditation and professional certification organizations to pursue evidence-based studies to support any surmised positive effects on organizational outcomes.

Although current leaders and management experts may presume to know the appropriate competencies and methods required to guide healthcare organizations, the persistent cost, quality, and access issues that permeate the industry erode perceptions of their effectiveness. As the United States healthcare system continues to contend with rising costs and persistent quality issues, leaders at all levels are seeking evidence-based factors that support improved outcomes. The institutional isomorphism theoretical model suggests that, in many cases, healthcare leaders pass along their own background and preparation as a professional model to emulate to students, staff, and direct reports. In the field of healthcare management, both CAHME-affiliated program graduates and ACHE affiliation are held in high regard and are often considered to be a prerequisite to a successful healthcare management career. The question is what evidence supports the continued proliferation of this idea in the contemporary operational environment? Our research may be highlighting the possibility that certification and accreditation standards foster groupthink within the industry [29].

Our scrutiny of both organizations should not be regarded as being dismissive of the potential value each brings to the industry. CAHME and ACHE both are well known for their support of competency-based education, which has led both academic programs and professionals to progressively advance in their understanding of key management principles [13,14]. Both organizations and multiple authors have published extensively on the topic of competency identification, development, and assessment. This makes our results more troubling and leads us to believe that the efforts to develop and support competency-based instruction may not be as effective as hoped. Based upon the analysis of the data for this manuscript, we do not currently find evidence to suggest that CAHME accreditation or ACHE affiliation have any direct effect on cost, quality, or access. We suggest that additional research in this area is warranted, but we also recommend that both CAHME and ACHE link their educational programming to healthcare setting performance outcomes. We believe that CAHME program graduates’ organizational impact needs to be quantitatively measured. Additionally, on the part of ACHE, the educational programming should be tailored to individual performance on the ACHE Fellow exam and should be focused based on those areas where a specific Fellow performed the most poorly.

## 8. Limitations and Future Research

The analysis was limited to both CAHME and ACHE affiliation of the senior leaders of healthcare organizations in the State of Texas. In addition, the analysis did not examine other accrediting agencies used in healthcare management education, which include the Association to Advance Collegiate Schools of Business (AACSB) and the Council on Education for Public Health (CePH). There are additional professional associations in the field of healthcare management, to include the Healthcare Financial Management Association (HFMA), the Medical Group Management Association (MGMA), and the Health Care Administrators Association (HCAA).

The focus of future research would be expanding the analysis to other relevant healthcare entities within the U.S. and including any accrediting agency or professional organization prevalent in the healthcare management education and the field of healthcare administrators. In addition, future work will include panel series analysis, which might help explain the effect of TMT certifications over time, if one exists at all.

Moreover, this study is limited in terms of the sample of hospitals because all hospitals in our study sample were located in Texas; therefore, our findings may not be generalizable to all U.S. hospitals. Future studies using hospital samples from all 50 states are needed to corroborate our findings.

## 9. Conclusions

Ultimately, this study determined that there is no significant relationship currently existing between healthcare leaders with accredited MHA graduate degree preparations and/or professional development initiatives with ACHE and their respective healthcare organization’s measurable performance outcomes. These results call into question if the present methods of senior leader preparation are sufficient to meaningfully move the needle on organizational performance. After decades of a lack of sustained meaningful improvement in cost containment, quality, and access in the industry, as claimed by its own IOM report, perhaps it is time that higher education degree accrediting organizations and leader professional development initiatives with professional development stakeholder organizations increase efforts to be more involved in their stakeholders’ longitudinal leadership development.

## Figures and Tables

**Figure 1 healthcare-09-00887-f001:**
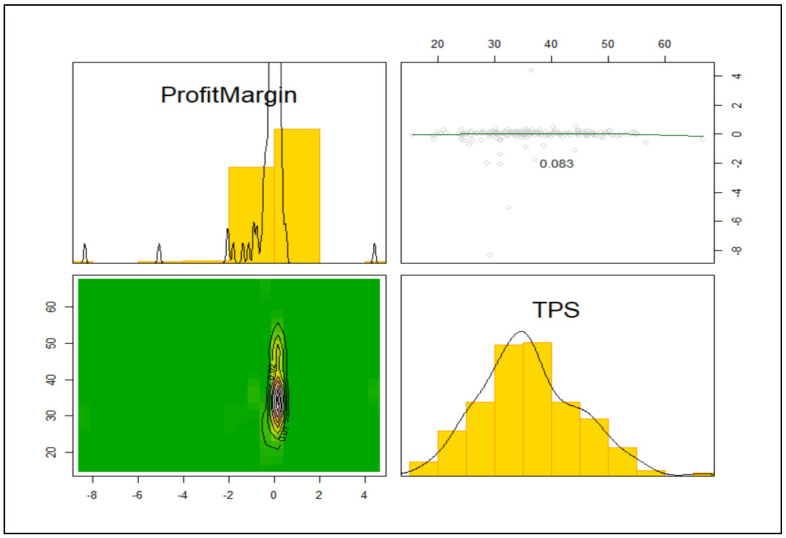
Scatterplot, bivariate plot, and univariate plots of the dependent variables.

**Figure 2 healthcare-09-00887-f002:**
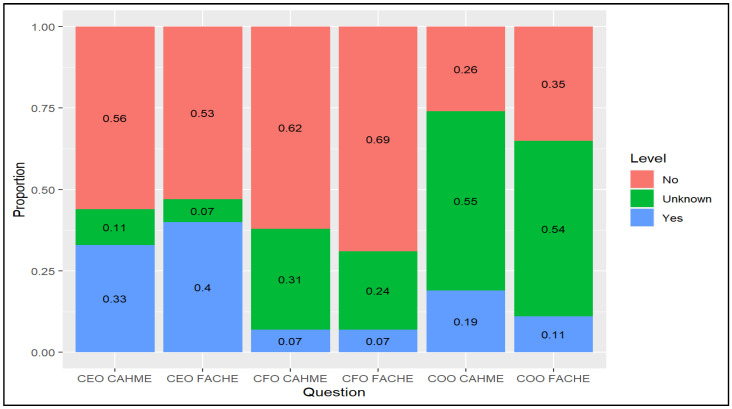
Status of CAHME/FACHE.

**Figure 3 healthcare-09-00887-f003:**
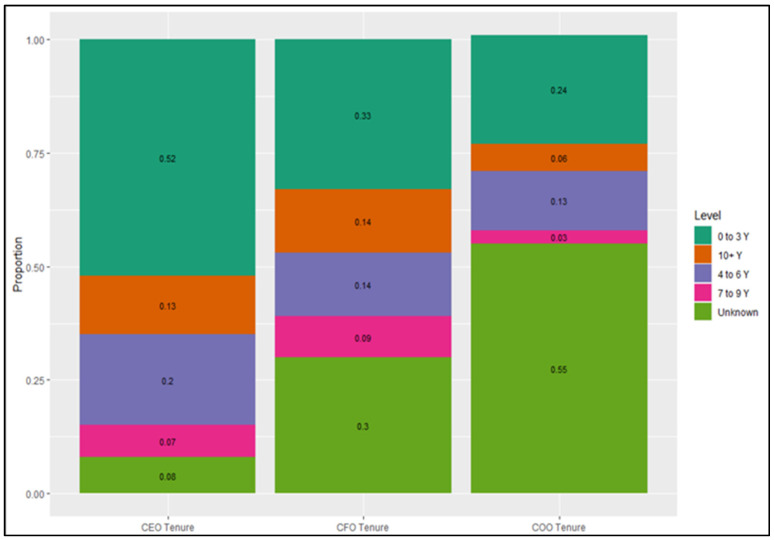
Tenure status for C-Staff.

**Table 1 healthcare-09-00887-t001:** Variables in the study and their associated sources.

Variables	Definition	Type	Source	Reason
Total Performance Score	Weighted Linear Combination of Subordinate Metrics	Quantitative	Centers for Medicare & Medicaid Services	
Net Operating Profit Margin	Operating Profit/Net Sales	Quantitative	Definitive Healthcare	
2020 Population Estimate	City of Hospital Population	Quantitative	U.S. Census Bureau	Demographic Control
2020 Population Density	Population/square meter	Quantitative	U.S. Census Bureau	Demographic Control
Native American	County % Native American	Quantitative	U.S. Census Bureau	Demographic Control
Black/African American	County % African American	Quantitative	U.S. Census Bureau	Demographic Control
Proportion over 65	% over 65	Quantitative	U.S. Census Bureau	Demographic Control
Unemployment, 2019	County Unemployment Rate	Quantitative	U.S. Census Bureau (BLS)	Economic Control
Household Income, 2018	Median County Income	Quantitative	U.S. Census Bureau	Economic Control
Adult Obesity	Rate/100 K	Quantitative	Centers for Disease Prevention and Control	Health Status Control
Cancer	Rate/100 K	Quantitative	Centers for Disease Prevention and Control	Health Status Control
COPD	Rate/100 K	Quantitative	Centers for Disease Prevention and Control	Health Status Control
Diabetes	Rate/100 K	Quantitative	Centers for Disease Prevention and Control	Health Status Control
Heart Failure	Rate/100 K	Quantitative	Centers for Disease Prevention and Control	Health Status Control
Case Mix Index	Complexity Adjustment	Quantitative	Centers for Disease Prevention and Control	Case Complexity Control
CEO Tenure	What is the CEO’s tenure?	{<=3, 4–6, 7–9, 10–12, 13–16, 16+, unknown} years	Secondary Source Mining (Hospital Websites, Linked-In)	Experience Control
COO Tenure	What is the COO’s tenure?	{<=3, 4–6, 7–9, 10–12, 13–16, 16+, unknown} years	Secondary Source Mining (Hospital Websites, Linked-In)	Experience Control
CFO Tenure	What is the CFO’s tenure?	{<=3, 4–6, 7–9, 10–12, 13–16, 16+, unknown} years	Secondary Source Mining (Hospital Websites, Linked-In)	Experience Control
CEO CAHME Status	CEO from CAHME-accredited university?	Qualitative, {No, Yes, Unknown}	Secondary Source Mining (Hospital Websites, Linked-In)	Predictor of Interest
COO CAHME Status	COO from CAHME-accredited university?	Qualitative, {No, Yes, Unknown}	Secondary Source Mining (Hospital Websites, LinkedIn)	Predictor of Interest
CFO CAHME Status	CFO from CAHME-accredited university?	Qualitative, {No, Yes, Unknown}	Secondary Source Mining (Hospital Websites, LinkedIn)	Predictor of Interest
CEO FACHE Status	CEO holds FACHE?	Qualitative, {No, Yes, Unknown}	Secondary Source Mining (Hospital Websites, LinkedIn)	Predictor of Interest
COO FACHE Status	COO holds FACHE?	Qualitative, {No, Yes, Unknown}	Secondary Source Mining (Hospital Websites, LinkedIn)	Predictor of Interest
CFO FACHE Status	CFO holds FACHE?	Qualitative, {No, Yes, Unknown}	Secondary Source Mining (Hospital Websites, LinkedIn)	Predictor of Interest
Do any of the CEO, COO, CFO have credentials/come from CAHME programs?	Recoded response based on previous 6 questions	Qualitative, {No, Yes}	Feature Engineering	Predictor of Interest

**Table 2 healthcare-09-00887-t002:** Descriptive statistics and quantitative variables.

Variable	Mean	SD	Median	Minimum	Maximum
Profit Margin	−0.086	0.830	0.031	−8.349	4.424
TPS	36.061	8.614	35.500	15.500	66.670
County Population	1,196,559.633	1,493,919.487	423,163.000	7306.000	4,713,325.000
Population Density	22,299.124	38,486.127	5453.706	856.992	246,914.129
Native American	0.003	0.002	0.002	0.000	0.014
Hispanic American	0.375	0.215	0.339	0.066	0.991
African American	0.111	0.075	0.094	0.000	0.335
Age 65+	0.135	0.035	0.122	0.095	0.301
Unemployment	3.570	0.940	3.300	2.100	9.800
Income	59,678.804	15,568.709	59,838.000	30,490.000	102,858.000
Adult Obese/100 K	31.117	4.643	30.000	21.800	47.300
Cancer/100 K	7.582	0.923	7.701	4.376	9.088
COPD/100 K	11.447	2.634	10.760	7.286	18.890
Diabetes/100 K	29.544	4.853	28.681	19.646	47.152
Heart Failure/100 K	15.855	2.871	15.589	10.345	27.965
Case Mix Index	1.737	0.308	1.817	0.980	2.234
Any FACHE Certification	0.503	0.510	1.000	0.000	1.000
Any CAHME Certification	0.492	0.501	0.000	0.000	1.000
Any Certification	0.764	0.426	1.000	0.000	1.000

**Table 3 healthcare-09-00887-t003:** Coefficient estimates for scaled Total Performance Score.

Variable	Estimate	Std. Error	t Value	Pr (>|t|)
Native	−0.173	0.069	−2.509	0.013
Population Density	0.163	0.070	2.341	0.020
Any Certification	−0.225	0.069	−3.273	0.001

## Data Availability

Publicly available datasets were partially compiled with the assistance of the Definitive Healthcare website (found here: https://www.defhc.com/). The primary publicly available datasets utilized were the Center for Medicare and Medicaid Services (CMS) Medicare Cost Reports (found here: https://www.cms.gov/Research-Statistics-Data-and-Systems/Downloadable-Public-Use-Files/Cost-Reports), the US Census Bureau (found here: https://www.census.gov/data/datasets.html), and the United States Department of Labor Bureau of Labor Statistics (found here: https://www.bls.gov/data/). Additional executive leadership level data were compiled through data mining the LinkedIn professional social network site (found here: https://www.linkedin.com/) and individual organization website research. All datasets were initially accessed on 27 August 2020.

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
