# Peer review of "Accreditation and Certification: Do They Improve Hospital Financial and Quality Performance?"

_healthcare, 2021, doi:10.3390/healthcare9070887_

Round 1
Reviewer 1 Report
This manuscript is very interesting and it is about a study that looks at accreditation and certification and whether or not there are impacts on performance outcomes in healthcare organizations (no effects were found in a sample of hospitals from the state of Texas). The subject matter is important. The manuscript is also written and articulated well. I feel that the manuscript can be improved a bit with some additional suggestions noted below:
- Lines 9 to 11 - How is organizational accreditation and certification impactful to academic programs and affiliates? Although you suggest this later, it is not clear in the abstract.
- Lines 36 to 38 - "Frustration levels..." sounds awkward, perhaps rephrase. Here is an example: - "The level of frustration among patients and clinicians has probably never been higher."
- Lines 66 to 68 - This review needs to be corroborated with a citation.
- Line 164 - Perhaps explain Ivy League vs. state schools as this is an international journal and not everyone would be familiar with these different educational institutions. E.g. elitism, level of endowments, etc.
- Some acronyms may not have been explained, please check MHA; especially since undergraduate and graduate programs were compared at one point.
If you decide to incorporate these revisions, please upload a manuscript that contains tracked changes or other method to highlight revisions. Thank you for the opportunity to review this work.
Reviewer 2 Report
Thank you to Healthcare for the possibility of the article review and compliments to the authors for their research effort.
After reading the article, my conclusion is that the paper can be accepted after minor revisions.
In the following notes, I summarize my assessments.
Referring to the Institutional Theory, the study explores the relationships between accreditation and professional certification and measurable organizational outcomes in some Texans Hospitals.
In particular, the authors test two research hypotheses:
"Hypothesis 1: Hospitals who employ health care executives who graduated from CAHME accredited programs and are ACHE certified demonstrate higher financial performance compared with hospitals who employ health care executives who did not graduate from CAHME accredited programs and are not ACHE certified.
Hypothesis 2: Hospitals who employ health care executives from CAHME accredited programs and are ACHE certified provide higher quality care compared with hospitals who employ health care executives who did not graduate from CAHME accredited programs and are not ACHE certified". (page 5, lines 256-263).
In my opinion, the article is well structured and consistent.
The authors expose the relevant literature and the theoretical reference model of their study (Isomorphism based on Institutional Theory).
The materials and methods (data, dependent variables, independent variables, statistical models) are clearly exposed.
The authors comment on findings adequately, even if they could better organize the theoretical and practical implications in the final part of the paper.
They also expose the limits of the article.
Minor concerns
To provide suggestions for possible improvements, I indicate below my minor concerns:
- the discussion of the results of the models related to Operating Profit Margin (page 11) could be supported with some detailed tables;
- The conclusions section is very short, and could therefore be developed better. Alternatively, the authors could combine the "Discussion" and "Conclusion" sections into a single paragraph (Discussion and conclusion);
- The considerations regarding "Limitations & Future Research" should not be indicated in a specific paragraph exposed before the conclusions, but should be placed at the end of the article (new section "Discussion and conclusions").
I don't feel qualified to judge the English language and style.
